# Knowledge of Sexually Transmitted Infections and HIV among People Living with HIV: Should We Be Concerned?

**DOI:** 10.3390/healthcare12040417

**Published:** 2024-02-06

**Authors:** Agnese Colpani, Andrea De Vito, Beatrice Zauli, Barbara Menzaghi, Andrea Calcagno, Benedetto Maurizio Celesia, Manuela Ceccarelli, Giuseppe Nunnari, Giuseppe Vittorio De Socio, Antonio Di Biagio, Nicola Leoni, Goffredo Angioni, Simona Di Giambenedetto, Gabriella D’Ettorre, Sergio Babudieri, Giordano Madeddu

**Affiliations:** 1Unit of Infectious Disease, Department of Medicine, Surgery and Pharmacy, University of Sassari, 07100 Sassari, Italy; 2Division of Infectious Diseases, “Ospedale di Circolo”, 21052 Busto Arsizio, Italy; 3Unit of Infectious Diseases, Department of Medical Sciences, University of Turin, 10123 Torino, Italy; 4Unit of Infectious Diseases, Department of Clinical and Experimental Medicine, University of Catania, 95125 Catania, Italy; 5Unit of Infectious Diseases, Department of Clinical and Experimental Medicine, University of Messina, 98122 Messina, Italy; 6Unit of Infectious Diseases, Santa Maria Hospital, 06129 Perugia, Italy; giuseppedesocio@yahoo.it; 7Infectious Diseases Clinic, IRCCS Policlinico San Martino Hospital, 16132 Genoa, Italy; 8Unit of Infectious Disease, SS Trinità Hospital, 09121 Cagliari, Italy; 9Infectious Diseases Unit, Catholic University of Sacred Heart, Fondazione Policlinico Universitario A. Gemelli IRCCS, 00168 Rome, Italy; 10Department of Public Health and Infectious Diseases, University of Rome Sapienza, 00185 Rome, Italy

**Keywords:** survey, HIV, STI, knowledge, U=U, education, prevention, people living with HIV, PLHIV, disclosure

## Abstract

Poor knowledge of sexually transmitted infections (STIs) and HIV among people with HIV (PLHIV) could worsen life quality. We aimed to investigate their STI and HIV knowledge, disclosure and undetectable = untransmittable (U=U). We proposed an anonymous questionnaire regarding STI and HIV to PLHIV attending ten Italian outpatient infectious diseases clinics. Moreover, disclosure and U=U were investigated. The calculated sample size was 178 people. Considering a missing response of 10%, the final sample size was 196. We enrolled 200 PLHIV (73.5% males), with a median age of 52.5 (IQR 41–59) years. The mean score was 7.61 ± 1.22 with no difference by gender, education, and employment. Significant statistical difference was observed by sexual orientation; bisexuals and those who preferred not to answer had a lower score than heterosexuals and MSM (*p* = 0.0032). PLHIV showed poor knowledge about HIV transmission (25% appropriately answered). Nearly 30% responded that virologically suppressed PLHIV could transmit the infection. Finally, 137 (68.5%) and 158 (79.0%) disclosed to the general practitioner and family and friends, respectively. Nearly 52.0% knew the meaning of U=U, and 83.6% highlighted its positive rebound. In conclusion, important knowledge gaps are present among PLHIV regarding U=U, and its implications are little-known. Improving PLHIVs’ awareness will undermine self-stigma and enhance life quality.

## 1. Introduction

Since the introduction of effective and safe therapies, the long-term quality of life for people living with HIV (PLHIV) has become one of the most significant concerns for infectious diseases specialists [1,2,3]. At the 20th International AIDS Conference in Melbourne, Australia, in 2017, UNAIDS launched ambitious global targets for HIV control to be achieved by 2020. These targets are widely known as “the three 90s”, advocating for 90% of PLHIV to be diagnosed, of which 90% should receive antiretroviral therapy, and 90% of those treated should achieve sustained virological suppression. There was wide heterogeneity among countries in reaching these targets [4]. Additionally, some authors proposed that the 90–90–90 targets were insufficient for a chronic condition like HIV, suggesting a fourth 90 addressing quality of life. In its latest report, UNAIDS raised the targets to three 95% by 2030 and added the three 10% to be reached by 2025. These 10% targets aim to ensure that less than 10% of PLHIV suffer from gender inequality, physical or sexual violence, stigma, discrimination, and lack of legal services, thereby improving PLHIV’s quality of life [5].

Improving quality of life should involve a comprehensive approach, including educating patients about their virological status and preventing other sexually transmitted infections (STIs). While HIV is decreasing in Western Europe, including Italy, STIs are increasing [6,7]. The risk of overlooking the health literacy of long-term patients exists; therefore, significant attention should be paid to educating people who regularly attend HIV services. Also, improving communication is crucial to prevent dangerous misunderstandings. Few data are available regarding STI proficiency among PLHIV. A recent survey in the general population reported at-risk behaviors linked to poor knowledge about STI transmission and prevention [8]. As the undetectable = untransmittable (U=U) message spreads, it is essential to reaffirm the risks associated with condomless sex and encourage routine STI screening for at-risk individuals.

Patients must understand that being virologically suppressed allows condomless sex without spreading HIV, but it does not protect against other STIs. Identifying at-risk behaviors and educating patients on this matter is crucial. It is essential to identify at-risk behaviors and educate patients in this regard. Education should also include a constant update regarding HIV. U=U promotes the finding that PLHIV with an HIV-RNA < 200 copies/mL cannot transmit the infection. This finding was well demonstrated by an analysis of the HIV Prevention Trials Network (HPTN) 052 trial, which showed that antiretroviral therapy prevented more than 96% of HIV infections in serodiscordant couples [9]; then, a 0% risk of HIV transmission for virological suppressed PLHIV in serodiscordant heterosexual and gay couples was confirmed by PARTNER 1 and 2 and The opposites Attract trials [10,11]. Several studies have demonstrated that the U=U message scale-up will reduce stigma and self-stigma and improve treatment adherence [12,13,14,15,16]. A recent article from the Icona foundation cohort showed that among 8241 PLHIV reaching the definition of U-status, nearly 97% of follow-up time over ten years has been spent with an HIV-RNA ≤ 200 cp/mL. This finding confirms the long-lasting virological suppression among most patients; thus, it can dispel any concern regarding the accountability of U=U [17].

A recent systematic review by Bor et al. assessed knowledge, awareness, attitudes, acceptability, and impact of U=U and treatment as prevention worldwide. Despite increasing awareness and a better understanding among PLHIV compared to people without HIV or unknown HIV status, both groups showed significant knowledge gaps, especially regarding transmissibility with undetectable viremia [18]. This message should be emphasized in HIV services, as U=U is well established, and PLHIV should be reassured about it.

The World Health Organization defines quality of life as “an individual’s perception of their position in life in the context of the culture and value systems in which they live and in relation to their goals, expectations, standards, and concerns” [19]. Disclosure has been shown to be part of this balance for PLHIV, associated with better treatment adherence, follow-up, and improved quality of life [20,21]. Disclosure to other health providers, family, and friends may face various barriers; these should be identified and addressed. Patients should be encouraged and supported in sharing their HIV status due to its beneficial impact on their lives.

In Italy, 1770 new diagnoses were reported in 2020; UNAIDS estimates around 140,000 PLHIV in the country in 2022. PLHIV aware of their serostatus have access to free counseling and care. Once engaged in the health system, they typically meet an infectious diseases specialist twice per year. Therefore, PLHIV are likely more aware than the general population regarding HIV, STIs, and U=U, being in constant contact with healthcare professionals. Additionally, many associations and NGOs provide continuous support and education to PLHIV and the general population.

We aimed to investigate the knowledge of STI prevention and transmission, at-risk behaviors, awareness regarding living with HIV and the U=U campaign, and disclosure attitude among PLHIV attending ten infectious diseases units in Italy.

## 2. Materials and Methods

We proposed an anonymous questionnaire to PLHIV attending ten outpatient clinics for infectious diseases in Italy. All PLHIV who were willing to participate in the survey were included. The survey comprised multiple-choice questions on STI and HIV transmission and prevention, as well as the meaning of U=U. We scored the responses by assigning 1 point for correct answers, 0.5 points for partially correct answers, and 0 points for incorrect answers, with a maximum achievable score of ten. The survey also included behavioral questions regarding condom use, as well as items related to disclosure to general practitioners, relatives, and friends. An open-ended question about the impact of U=U on self-perception concluded the questionnaire (Appendix A). The virological status of participants was not investigated, nor were any questions about the use of pre-exposure prophylaxis (PrEP) by partners included in the survey.

Two infectious diseases residents (A.D.V., A.C.) formulated the questions, which were then revised by a senior infectious diseases Professor (G.M.). These items had previously been used in a survey targeting the general population, with additional questions on disclosure and the impact of U=U included for this particular study.

Data on participants’ age, sex, regional origin, sexual orientation, education level, and occupation were collected.

The research was conducted in accordance with the Declaration of Helsinki [22]. This study did not involve clinical trials, and all participant’s data were completely deidentified. In compliance with current national legislation from the Italian Medicines Agency, formal consent was not required for this type of study. Ethical approval for this research was obtained from the Ethical Committee of Sardinia (code: QUEST-HIV).

Assuming an estimated knowledge of 25% among the general population [23] and a presumed 45% among PLHIV, based on an alpha error of 0.05 and a beta error of 0.20, the sample size should be 178. Considering a missing response of 10%, final sample size was 196 subjects. Nonetheless, all answers were accepted even beyond the required minimum.

Before performing the statistical analysis, data distribution was evaluated with the Kolmogorov–Smirnov test. A *p* > 0.05 was considered as normality of the distribution. Then, data were elaborated as numbers on total (percentages) and mean ± standard deviation. Next, continuous variables with parametric distribution were compared with Student’s *t*-test or with one-way ANOVA. Categorical variables were evaluated with the Pearson chi-squared test. We performed a linear regression analysis to assess relationship to between the score and the different variables.

The statistical significance level was established as *p* < 0.05. The Stata statistical software package, version 16.1 (StataCorp LP, College Station, TX, USA), was used for data processing and statistical analysis.

## 3. Results

We collected 200 questionnaires from PLHIV. All participants were regularly attending infectious diseases units and were on antiretroviral treatment. Each participant was infected with HIV virus type 1. The median age of the participants was 52.5 years, with an interquartile range (IQR) of 41–59 years. The majority of the respondents were male (N = 147, 73.5%). In terms of sexual orientation, 134 participants (67.0%) identified as heterosexual, 74 (37.0%) as men who have sex with men (MSM), 24 (12.0%) as bisexual, and 8 (4.0%) chose not to disclose their sexual orientation. The predominant educational level was a high school diploma, and most participants were employed. The sociodemographic characteristics of the participants are detailed in Table 1.

The mean score obtained in the questionnaire was 7.61 ± 1.22, with ten as the maximum score possible. Analysis revealed no significant differences in scores based on age, gender, educational level, or employment status. However, a notable statistical difference emerged when comparing scores by sexual orientation. Participants who identified as bisexual and those who did not disclose their sexual orientation had lower scores compared to those identifying as heterosexuals and MSM (*p* = 0.0032). This finding was further corroborated by linear regression analysis, which indicated an R-squared value of 0.0661 and a *p*-value of 0.038, suggesting a modest but significant influence of sexual orientation on questionnaire scores.

Overall, the survey indicated a significant lack of knowledge among people living with HIV (PLHIV) about HIV transmission routes, as only 52 participants (26.0%) provided adequate answers. Despite 188 respondents (94.0%) correctly stating that living with a PLHIV is not dangerous, 46 patients (23.0%) incorrectly believed that PLHIV could transmit the infection even when on antiretroviral therapy and virologically suppressed. Furthermore, 13 participants (6.5%) were unsure if PLHIV on antiretroviral treatment and virologically suppressed could transmit the infection.

A particularly concerning finding was related to misconceptions about contraceptive pills. Twenty-six participants (13.0%) either incorrectly believed that contraceptive pills protect against sexually transmitted infections (STIs) or did not know the answer. Additionally, nearly 10% of respondents had misconceptions about effective prevention methods; a common error was the belief that proper intimate hygiene alone could prevent STIs. Moreover, over 50 respondents believed that only specific types of sexual intercourse carry the risk of STI transmission, such as receptive anal sex exclusively (Figure 1).

In terms of behavior, the survey showed that 19 participants (9.5%) never use condoms during casual sexual intercourse, and 37 (18.5%) rarely do, while only 57 (28.5%) always use condoms.

Regarding their understanding of undetectable = untransmittable (U=U), 18 respondents (9%) mistakenly believed that it implies that a PLHIV cannot transmit any STI. Nearly 52.0% correctly understood the meaning of U=U, and 87 participants (43.5%) reported that this knowledge changed their self-perception and had a positive impact on their relationships, sexual life, and expectations around parenthood. Responses regarding the impact of U=U were summarized into key statements reflecting the various domains it affected: “It gave me more peace and less fear”; “It gave me awareness”; “When they told me U=U also applied to maternity and labor, I felt less untouchable”; “It gave me the courage to start a long-term relationship”; “It made me understand that I could have a normal life”; “I eventually accepted my status”; “It gave me hope and self-confidence”; “I feel safe and free in my sexual life, I no longer feel guilty”.

In terms of disclosure, only 137 participants (68.5%) reported having discussed their HIV status with their general practitioner. The primary reasons for not disclosing included the perceived lack of necessity, shame, and fear of judgment. However, a higher number of participants (158 (79.0%)) disclosed their status to at least one person among family, friends, or partners, with nondisclosure often attributed to fear of judgment, fear of being avoided, and perceived lack of necessity. The specific reasons for disclosure and nondisclosure are detailed in Figure 2.

Finally, 181 (90.5%) looked for more informational campaigns regarding STI and HIV, and 163 (81.5%) declared they had already learned something new or felt pushed to collect further information on the matter by filling out the present questionnaire.

## 4. Discussion

Scarce research regarding PLHIV knowledge of STI, HIV, U=U, and attitude towards disclosure is available. A poor understanding of STI and HIV prevention can lead to at-risk behaviors, such as condomless sex with occasional partners. Regarding U=U and the risk of HIV transmission, misbeliefs can enhance self-stigma and disrupt the quality of life [13]. Addressing these gaps should be a priority to improve life quality and self-awareness. With regard to disclosure, it is associated with improved adherence to treatment and a better outcome [20,21]. Understanding the reasons for nondisclosure is crucial to encourage PLHIV to disclose eventually. In our study, PLHIV showed adequate global knowledge of HIV and STI. However, significant shortcomings were recorded. Considering that this is a selected population of people regularly attending infectious diseases clinics and accepting to participate in such interventions, we could have hoped for better scoring. Physicians have the unique opportunity of regularly counselling the patients and should not underestimate continuous education as a valuable tool to prevent STI and improve quality of life; long-term patients should not be left out.

A study aiming to investigate the knowledge of STI and HIV among the general population through the same ten items reported a similar mean score to the one registered among PLHIV (7.62 ± 1.42 and 7.61 ± 1.22). However, participants from the two studies are not matched by age, sex, sexual orientation, level of education, and occupational status; thus, the results are difficult to compare [8]. An anonymous online survey conducted among men who have sex with men in England highlighted a better understanding of HIV and STI among PLHIV compared to people not living with HIV or with unknown HIV status. The study included 3663 participants, of which 3157 were HIV negative or with unknown HIV status and 489 PLHIV. The interview had 11 items on STI; one point was assigned to correct answers, and a score inferior to six was considered “poor knowledge”; the proportion of respondents with “poor knowledge” was higher among people without HIV or with unknown HIV status. The median score was seven among PLHIV and six among people without HIV or with unknown HIV status, with a statistically significant difference between the two groups. Items differed from our survey; however, this study confirms that PLHIV may be better informed than their counterparts, but they still display significant knowledge gaps, as does the general population [24].

Regarding U=U, few studies report knowledge of it among PLHIV. A web-based, self-reported, cross-sectional survey investigating the exposure to the U=U message was conducted in 25 countries between 2019 and 2020. Overall, 2389 PLHIV were involved, of which 1588 (66.5%) discussed U=U with their healthcare provider. A slightly higher percentage was reported from Europe (68.4%); however, data from Italy were not available in the manuscript [25]. Our study shows a much worse picture of knowledge of the U=U message, while better than what was reported among the general population by De Vito et al. In the study mentioned above, only 16.3% of participants answered that they knew the U=U campaign (vs. 52.0% in the present investigation) [8]. Both percentages should be improved, as many studies report how knowing U=U meaning undermines both stigma and self-stigma [12]. In addition, 18 participants believed they knew the meaning of U=U, while they gave a wrong answer when asked about it; assuming that U=U means that no STI can be transmitted can be very dangerous, preventing the patient from regular screening for STIs other than HIV. Therefore, improving patient–doctor communication and carving out enough time to discuss these topics during the visits is crucial. While the best way to reach the general population is still to discuss, informing people constantly attending health services should be much easier. It is up to the healthcare provider to ensure proper and updated education for PLHIV.

As for the impact of U=U, 87 (43.5%) answered that U=U changed their life; however, only 72 (36%) responded to the open-answer question specifying which was the impact on themselves and their social life. As easily understandable, many said that the U=U message brought less fear of transmitting the infection and more freedom in sexual intercourse; also, improved confidence in engaging in stable relationships was mentioned. In addition, participants felt safe in designing a family. Furthermore, enhanced self-esteem and peace of mind were registered. On the contrary, 68 (34%) declared that U=U had no impact on self-perception or their life; among these, only 28 correctly answered regarding the meaning of U=U. A reason for this discrepancy is suggested by a study conducted in Australia; in this study, the authors hypothesized that consensus regarding U=U among PLHIV may mask a lack of confidence in real-life choices [26]. Other studies confirm that U=U uptake reduces self-stigma and improves the quality of life [13]. Thus, more effort should be placed on reaffirming the accountability of U=U with each patient coming to visit.

As for disclosure, few people have talked to their general practitioner about the infection. Reasons for nondisclosure included no need, shame, and fear of judgment. The latter is particularly alarming and should be addressed through educational programs targeting general practitioners and noninfectious diseases specialists. This hesitancy is undoubtedly due to perceived stigma, and this behavior could lead to dangerous consequences, such as unexpected drug–drug interactions. A study in the Netherlands shows how disclosure perspectives differ completely between PLHIV and healthcare providers. The latter reported that due to their confidentiality duty, they did not understand disclosure concerns. Also, they answered that they should know their patients’ serostatus to reduce occupational risk and adopt preventive measures. It should be highlighted that the study was conducted in 2016 when the U=U campaign was beginning.

On the contrary, PLHIV declared that disclosure was a courtesy not due to their healthcare provider [27].

This study confirms that many health workers may not be aware of their stigmatizing behaviors. Also, investigating if the U=U campaign modified physicians’ attitudes somehow would be interesting. It would be compelling to know if the U=U impacted not only PLHIV self-perception but also health workers’ attitudes towards PLHIV. The U=U campaign can help defeat stigma against PLHIV; less stigma means reducing barriers to health access in the sanitary context.

A slightly higher number of participants disclosed to at least one person among family and friends. However, the fear of being avoided and judged is still highly reported. A similar prevalence was reported by Endalamaw et al. in Ethiopia (76%) [28], while a lower rate of disclosure resulted from a systematic review conducted in China (65%) [29]; however, both studies considered disclosure to sexual partners only and come from very different contexts [28,29]. An observational study conducted in 2012 and 2013 aiming to assess disclosure attitudes and predictors among women living with HIV reported a high prevalence of disclosure (1724/1929, 89%); women from Western Europe showed better attitudes towards disclosure, data that we hope has improved since the spreading of the U=U campaign [30]. Disclosure is long known to be associated with higher treatment adherence and improved quality of life [20,21,30]; however, targeted interventions to improve serostatus disclosure are lacking [31]. Continuous education and sensitization of health workers are crucial to avoid misbeliefs and stigmatizing behaviors. Also, psychological support and peer-to-peer education may improve PLHIVs’ confidence in disclosing. Assistance in sharing HIV diagnosis with relatives, friends, and partners should be offered to encourage this decision.

Regarding behavioural items, few participants declared a consistent use of condoms. A study conducted in Ethiopia with a similar sample size (352 participants) reported a similar trend, with 79.8% of participants engaging in at least one at-risk behavior. These data are in line with our previous study conducted among the general population, where 1.2% reported never using condoms, 15.3% referred to an occasional use, and only 30.5% said they always use condoms. Again, the two study groups are not matched by sociodemographic characteristics [8]. However, considering the present study population, constantly in contact with healthcare providers, behavioural changes could be easier to achieve and must be advocated. An interesting review investigating the effectiveness of psychological and behavioural interventions in reducing high-risk sexual behaviors confirmed a positive role of such interventions in condom use implementation; however, this increase seemed to peak right after the intervention and then declined over time [32]. Most of the studies reported in this review investigated condom use attitudes before the U=U era, which could have radically changed sex behaviors among PLHIV. Nonetheless, occasional condomless sex still represents a risk for STIs other than HIV in PLHIV; thus, educational interventions should be promoted.

Lastly, our participants support promoting more informational campaigns on HIV and STI and reported great utility from the present survey as an educational tool. The same finding was reported in our previous study among the general population [8].

Our study has some limitations; first, this is a selected sample of patients regularly attending healthcare provisions and accepting to participate in such interventions. These factors could lead to an overestimation of health literacy and awareness of PLHIV in general. We think that other kinds of interventions should be planned to address marginalized populations and better assess gaps in the knowledge of HIV and STI. Also, most participants came from Sardinia and Lazio, and many Italian regions were not represented. Italy is a heterogeneous country from a sanitary and educational point of view; thus, a better-distributed sample could lead to more generalizable results. We intend to involve other colleagues from other centers and regions to collect more data and more representative of the national situation. Furthermore, our population was mainly male. Women often carry a heavier burden than men; also, they are often left out of clinical trials, and, because they are outnumbered, are less represented in clinical studies [33]. Further studies should address women’s perspectives. Another limitation is that data regarding the year of diagnosis were not collected, which could affect HIV proficiency and awareness. Also, regarding behavioural items, only condom use was investigated.

In contrast, other prevention measures (e.g., pre-exposure prophylaxis (PrEP) and PrEP with doxycycline (doxyPrEP)) and other at-risk behaviors (e.g., the use of substances to enhance sexual experiences) were not evaluated. Also, doxyPrEP is not used in Italy. Thus, we decided to focus on condom use to assess the behavioural domain.

## 5. Conclusions

Our study highlights several critical areas that require attention in the care of people living with HIV (PLHIV). Firstly, intensifying the U=U campaign should be a priority, extending beyond the general public to include all PLHIV. It is crucial to ensure that the U=U message is consistently communicated to patients, with a clear explanation of its meaning. PLHIV need to be reassured about the reliability of an undetectable status, while also understanding that U=U specifically pertains to HIV and not to other STIs. Secondly, there is a need to promote behavioral changes through improved literacy about STIs. This involves not just disseminating information but also fostering a deeper understanding that can lead to safer practices. Thirdly, concerning disclosure, it is necessary to implement interventions targeting health professionals and relatives to encourage and support the disclosure of HIV serostatus and help normalize HIV infection. Considering that our study’s participants are a selected group, regularly attending HIV clinics and receiving initial counseling and biannual specialist visits, the modest outcomes of the survey suggest the need for further studies and initiatives. These should focus on enhancing the training and capacity of healthcare professionals in HIV and STI management.

Lastly, our findings indicate the importance of conducting further research involving marginalized populations who are less likely to participate in self-administered surveys. This will help to map the knowledge of HIV and STIs within these groups and facilitate the planning of tailored interventions that address their specific needs and circumstances.

## Figures and Tables

**Figure 1 healthcare-12-00417-f001:**
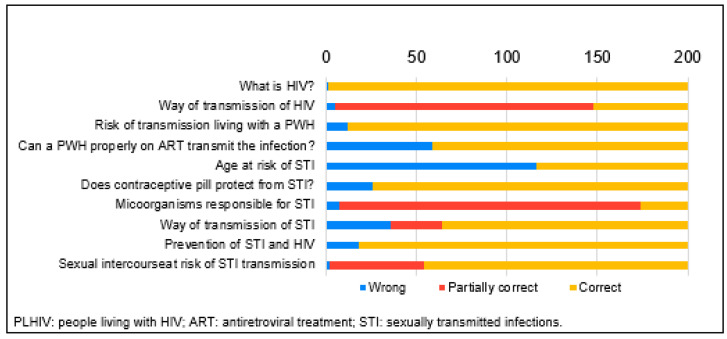
Answers from 200 participants.

**Figure 2 healthcare-12-00417-f002:**
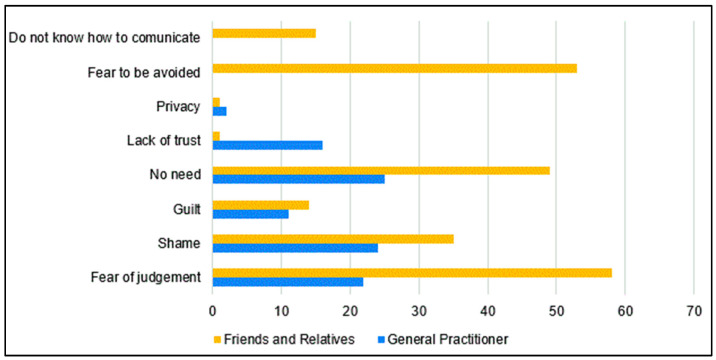
Reasons for nondisclosure to the general practitioner, family, and friends among 200 participants.

**Table 1 healthcare-12-00417-t001:** Sociodemographic characteristics of 200 participants.

Variables	Participants (n = 200)
Age (years), median (IQR)	52.5 (41–59)
Male gender at birth, n (%)	147 (73.5)
Sexual orientations, n (%)	
Heterosexual	94 (47)
Men who have sex with men	74 (37)
Bisexual	24 (12)
Not answered	8 (4)
Level of Education, n (%)	
None	5 (2.5)
Primary school	39 (19.5)
Secondary school	94 (47)
Bachelor’s degree or above	62 (31)
Occupation, n (%)	
Unemployed	36 (18)
Employee	93 (46.5)
Self-employee	37 (18.5)
Student	2 (1)
Retiree	32 (16)

IQR: interquartile range, n: number.

## Data Availability

Research data will be available upon request.

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
