# Peer review of "Knowledge of Sexually Transmitted Infections and HIV among People Living with HIV: Should We Be Concerned?"

_healthcare, 2024, doi:10.3390/healthcare12040417_

Round 1
Reviewer 1 Report
Comments and Suggestions for Authors
Dear Authors,
My compliments for addressing an important health issue in Italy.
The paper will need minor revisions before publication. I hope the suggestions in the attached file are useful for revising the paper.

Dear Edotor,
I found the paper well written but some minor errors are seen . Revision to correct / include some important facts and figures is also required. I expect the detailed comments provided to the authors will help the authors to add some newer perspective as well as address the existing factual issues. This of course does not include the review of statistical calculations.
Regards
Author Response
Thank you for your detailed revision, which has helped us improving our manuscript.
R: Line 23: Please check for the intended meaning.
AR: Unfortunately, there is not enough space in the abstract to better explain the meaning of the sentence. However, we think we provided enough clarification in the main text. If you have any suggestions on how to improve the abstract, we would be happy to consider it.
The outcome of STI knowledge of the participants is missing in the abstract. Try to ensure that the concluding line aligns with the relevant and important conclusion/s given in the main paper.
AR: Regarding the Abstract, I am not sure about the meaning of your comment. STIs knowledge was investigated through questions regarding transmission route and prevention. Our aim was to assess the level of awareness among a high-risk population, due to HIV coinfection and at-risk behaviours. This concept is better explained in the Discussion, due to the number of characters allowed in the abstract. Also, we have discussed how to improve this specific aspect.
R: Introduction: Adding the magnitude of the HIV problem in Italy in terms of facts and figures and the existing system of educating HIV patients may be considered by the authors. This will add value to the paper and make the study important from a public health point of view.
AR: Thank you for you for your comment. Regarding the Introduction, we have implemented as suggested.
R: Line 126: Was this study conducted in the same population as reference number 23? If yes, please mention in the text here and if not, then specify the population size to justify the epidemiological soundness of the sample size. Furthermore, please provide precise information on the total number of HIV positive patients attending the 10 infectious disease units and the number of patients selected for the study. What were the inclusion and exclusion criteria of the participants?
AR: Thank you for your insightful comments and queries regarding our study. in our study indeed references the findings from study number 23 to establish a baseline understanding of general population knowledge in Italy. However, our research was not conducted on the same population as that of reference 23. Instead, we utilized the data from reference 23, which assessed the knowledge of the general population in Italy, as a comparative baseline to hypothesize a presumed higher level of knowledge among People With HIV (PWH). This comparison was pivotal in setting the context for our study, which focused exclusively on PWH. We did not selectively choose patients for our study to minimize selection bias, ensuring a more representative sample of the PWH population. By not restricting participation to a predefined subgroup, we sought to provide a comprehensive overview of the knowledge level among all PWH attending these clinics.
As for the inclusion and exclusion criteria, our study included all people with HIV who were attending the outpatient clinics of the infectious disease units during the study period. The only exclusion criterion was the unwillingness or inability of patients to complete the questionnaire. This inclusive approach further supports our efforts to minimize selection bias.
R: Figures in Table 1 and the preceding text do not match for heterosexuals. In the text, it is 134 whereas in the Table it is mentioned as 94. Please re-check the statistical calculation regarding this variation in numbers.
Further, while saying most participants qualify, please give (%) for clarity.
AR: Thank you for your attention. There was a typo in the text. The heterosexual were 94.
R: Can you specify whether Italy has a system of counselling HIV patients? If yes, then the discussion needs to consider including counselling of HIV patients and if not, the role of introducing counselling, as done in some countries can be considered. This might add a new dimension and value to your study. Proposing KAP studies of the counsellors/ relevant health workers and their capacity building can also be a value add to the paper and may be considered under discussion and conclusions.
AR: As for the introduction, we have implemented it including your precious suggestions.
Reviewer 2 Report
Comments and Suggestions for Authors
This manuscript by Colpani A et.al investigates the knowledge of sexually transmitted infections as a part of HIV prevention and transmission, risk behaviors, awareness among the people living with HIV in Italy.
They enrolled 200 participants for this study. Participant were allowed to answer the questionaries in the form of multiple-choice questions related to STI and HIV transmission and prevention. The finding of this study shows that nearly half of the participants did not know the undetectable equals untransmittable. Therefore, this study concludes that Improving awareness among the PLHIV will undermine self-stigma and enhance life quality.
This study is conducted well, and Figure 1 is well presented.
The only concern is the quality of English language and some other typo.
thanks
Comments on the Quality of English Languagethe quality of english need to be improved
Author Response
This manuscript by Colpani A et.al investigates the knowledge of sexually transmitted infections as a part of HIV prevention and transmission, risk behaviors, awareness among the people living with HIV in Italy.
They enrolled 200 participants for this study. Participant were allowed to answer the questionaries in the form of multiple-choice questions related to STI and HIV transmission and prevention. The finding of this study shows that nearly half of the participants did not know the undetectable equals untransmittable. Therefore, this study concludes that Improving awareness among the PLHIV will undermine self-stigma and enhance life quality.
This study is conducted well, and Figure 1 is well presented.
The only concern is the quality of English language and some other typo.
Dear review, thank you for your feedback on our manuscript. We appreciate your positive comments and acknowledge the concerns regarding language and typographical errors. We will revise these aspects carefully to improve clarity and quality.
Reviewer 3 Report
Comments and Suggestions for Authors
The importance of increasing awareness about STIs and blood-borne pathogens is well-recognized in global programs. To enhance the novelty of this study, careful and comprehensive design is essential.
Following are the specific comments for addressal:
1. The chosen population for this study falls within the median age group of 52.5 years. However, considering the highly sexually active groups targeted for this study, a slightly younger age group could have been considered.
2. During the collection of demographic details, clinical information about existing or past STDs was not gathered. Including such information could have provided additional insights into individuals' awareness of STIs and their treatment.
3. The study's enrollment criteria should be explicitly stated.
4. The sociodemographic data table lacks clear information regarding the gender composition of the study population. It is important to know the percentage of women included, especially if the study focused on a specific gender. The author acknowledges bias towards men in the limitations section, indicating a missed opportunity to include a crucial study group and potential bias in representing generalized data.
5. The study appears to underestimate the importance of obtaining ethical approval from participants and securing their consent, which may not have been adequately addressed in the article. Please include the same during the revised submission.
6. In the Materials and Methods section, the author mentions the questionnaire used but fails to include a reference for it. Including the reference would enhance the transparency and replicability of the study.
7. The questionnaire used in the study could benefit from additional information on STI prevention and treatment, as well as the status of viral load in people living with HIV (PLHIV).
8. Gathering feedback from PLHIV before and after counseling regarding their knowledge of STIs and HIV could significantly impact the study's implementation strategies for awareness programs.
9. Exclude the repetition of texts in the manuscript and make it more precise and specific to enhance clarity of representation.
Author Response
The importance of increasing awareness about STIs and blood-borne pathogens is well-recognized in global programs. To enhance the novelty of this study, careful and comprehensive design is essential.
Following are the specific comments for addressal:
R: The chosen population for this study falls within the median age group of 52.5 years. However, considering the highly sexually active groups targeted for this study, a slightly younger age group could have been considered.
AR: The population was not selected by the authors. All patients living with HIV and willing to participate to the study were admitted.
R: During the collection of demographic details, clinical information about existing or past STDs was not gathered. Including such information could have provided additional insights into individuals' awareness of STIs and their treatment.
AR: We did not gather clinical information during the survey, and we were unable to retrieve them from medical records because the questionnaire was anonymous. However, it could be interesting to match such data with the results, and we will consider this suggestion for further studies.
R: The study's enrollment criteria should be explicitly stated.
AR: Thank you for your comment, we added a sentence in the Methods to clarify this point.
R: The sociodemographic data table lacks clear information regarding the gender composition of the study population. It is important to know the percentage of women included, especially if the study focused on a specific gender. The author acknowledges bias towards men in the limitations section, indicating a missed opportunity to include a crucial study group and potential bias in representing generalized data.
AR: Thank you for your comment. The Table includes number and percentage of male at birth. Which means that any other participants were a woman.
R: The study appears to underestimate the importance of obtaining ethical approval from participants and securing their consent, which may not have been adequately addressed in the article. Please include the same during the revised submission
AR: The study was approved by the local Ethic committee and conducted according to the Helsinki declaration, as reported in the Methods.
R: In the Materials and Methods section, the author mentions the questionnaire used but fails to include a reference for it. Including the reference would enhance the transparency and replicability of the study.
AR: We will provide the questionnaire as a supplementary material, thank you for noticing it.
R: The questionnaire used in the study could benefit from additional information on STI prevention and treatment, as well as the status of viral load in people living with HIV (PLHIV).
AR: Thank you for the suggestion, we will take it into consideration when conducting further surveys.
R: Gathering feedback from PLHIV before and after counseling regarding their knowledge of STIs and HIV could significantly impact the study's implementation strategies for awareness programs.
AR: Thank you for the suggestion. However, we think that this could be hard to assess. In Italy, PLHIV receive a counselling when undertaking a screening and when the results are positive. It would be critical to expose a newly diagnosed person to such a survey, being the moment of the diagnosis a delicate one. We think that targeting physicians and improving their awareness and capability of providing effective counselling, could be a more suitable way to improve patients’ knowledge.
R: Exclude the repetition of texts in the manuscript and make it more precise and specific to enhance clarity of representation.
AR: Thank you for the comment, we will assess the clearness of the paper carefully.
Round 2
Reviewer 3 Report
Comments and Suggestions for Authors
Thank you for addressing most of the raised concerns in the revised submission.
Author Response
Dear Reviewer,
thank you for your contribution in improving our Manuscript.